# Mortality Trends due to Falls in the Group of People in Early (65–74 Years) and Late (75+) Old Age in Poland in the Years 2000–2020

**DOI:** 10.3390/ijerph20065073

**Published:** 2023-03-14

**Authors:** Monika Burzyńska, Tomasz Kopiec, Małgorzata Pikala

**Affiliations:** 1Department of Epidemiology and Biostatistics, Medical University of Lodz, Żeligowskiego 7/9, 90-752 Lodz, Poland; 2Health Systems Development Department, Medical University of Lodz, Muszyńskiego 2, 90-752 Lodz, Poland

**Keywords:** falls, ageing, mortality trends, Poland

## Abstract

The aim of the study was to assess mortality trends due to falls in early (65–74 years) and late (75+) old age groups in Poland in 2000–2020. The study used a database of all deaths due to falls in two age groups. Per 100,000 men in early old age, the crude death rate (CDR) increased from 25.3 in 2000 to 25.9 in 2020. After 2012, a statistically significant decrease was observed (annual percentage change (APC) = −2.3%). Similar trends were noted for standardized death rates (SDR). Among men 75 years and older, the CDR values between the years 2000 and 2005 decreased (APC = −5.9%; *p* < 0.05), while after 2005, they increased (1.3%; *p* < 0.05). The SDR value decreased from 160.6 in 2000 to 118.1 in 2020. Among women aged 65–74, the CDRs values between 2000–2020 decreased from 13.9 and 8.2 per 100,000 women. The SDR value decreased from 14.0 to 8.3, respectively (2000–2007: APC = −7.2%; *p* < 0.05). Among women aged 75+, the CDR value decreased from 151.5 to 111.6 per 100,000 but after 2008, they began to increase (APC = 1.9%; *p* < 0.05). SDR decreased from 188.9 to 98.0 per 100,000 women. Further research on the mortality in falls is needed in order to implement preventive programs.

## 1. Introduction

Progressive ageing of the global population accompanied by an increase in the proportion of older people determines the current structure of health needs and challenges that healthcare systems and social welfare systems, as well as society as a whole, have to face. In Poland, between 2000 and 2020, the life expectancy of women and men increased by 2.7 and 2.9 years, respectively, and reached values of 80.7 and 72.6 [1], thus shaping the pattern of morbidity and mortality of the entire population. Needs of elderly population are very diverse, and this age group itself is heterogeneous. Internal diseases are their main health problems. In addition, a characteristic feature of old age is progressive multimorbidity, i.e., coexistence of several chronic diseases [2]. For the purpose of assessing this phenomenon, the concept of the so-called “major geriatric problems”, defined as “chronic disorders that gradually lead to functional incapacity and thus negatively affect the quality of life of older patients”, was introduced into modern gerontology [3]. These diseases include, in particular, sphincter incontinence, visual and hearing impairment, dementia disorders with delirium, old-age depression, iatrogenic syndrome, impaired mobility, and falls.

The World Health Organization (WHO) defines a fall as “an event which results in a person coming to rest inadvertently on the ground or floor or other lower level” [4]. Falls are a major global public health problem. It is estimated that 684,000 fatal falls occur each year, which makes them the second most common cause of death from unintentional injury after traffic accidents. All around the world, mortality from falls is highest among people over 65 years of age. In this age group, one person in three falls down at least once a year. In the group of people aged 70 years and older, the incidence of falls is 32–42%, while by the age of 80, a fall occurs in every second person [5,6]. It is worth noting that 30–60% of long-term care residents aged 65 years and older and 20% of hospitalized people in this age group experience falls at least once a year [7]. Risk factors of falls, according to the simplest classification, can be divided into intrinsic and extrinsic [8]. The first group includes nonmodifiable factors, such as age and gender and health-related factors, while the other group includes environmental factors such as home surroundings, government policy on environmental design, housing standards, public transport, neighborhood conditions, and social networks [9]. In the elderly population, an accumulation of risk factors can be observed in both groups. Intrinsic factors are exacerbated by the natural ageing process, developing medical conditions, and medication use. The most significant causes of falls in this group, which meet evidence-based medicine (EBM) criteria, include muscle weakness, history of falls, gait and balance disorders, use of gait-assist devices, visual impairment, joint inflammation, depression, memory disturbances and age over 80 years [10]. Individual predictive factors also include the so-called “post-fall syndrome”, which manifests with a fear of falling down again and results in reduced physical activity, which in turn is an important element of fall prevention [11,12,13]. External risk factors of falls include hazards in home environment, such as slippery surfaces, high doorsteps, too-bright or too-dim light, and cabinets which are fixed too high, as well as outside hazards, such as uneven surfaces or architectural barriers which prevent mobility [14]. Poor financial conditions, lack of social support, and limited availability of healthcare services are other causes of falls [15].

In 2020, the standardized death rate due to falls in people aged 65 years and older in Poland was 43.1 per 100,000 population in this age group, higher than the average value in European Union countries (40.9) and WHO European Region countries (33.9) [16].

In addition to being the most common cause of fatal injury among people over 65 years of age, falls are also a cause of nonfatal injuries leading to impaired functioning due to complications. Falls can cause hip and thigh injuries both in men and women. They are the most common reason for hip fracture hospital admissions (9 in 10 cases). Other injuries that result from falls include head injuries, wrist fractures, humerus, intracranial hematomas, and injuries to internal organs [17]. In the group of people aged 75 and older, 70% of falls are fatal. About 5% result in a fracture and 10–20% result in severe soft tissue injury [18]. Advanced age, frailty, and pre-existing medical conditions decrease the likelihood that older individuals will recover from fall-related injuries [19]. The Global Burden of Diseases (GBD), Injuries and Risk Factors Study 2017 shows that falls are ranked as the 18^th^ leading cause of age-standardized disability-adjusted life year (DALY) rates [20]. Between the years 2000 and 2019, DALYs due to falls in the elderly increased globally by 60% (18% per 100,000 population) [21], which confirms that this is an important and growing public health problem.

Changes in mortality due to falls among people over 65 years old were analyzed. The strength of the study is the completeness of death registration and the long period of the analysis. The results of studies of mortality of the elderly usually define this group as 65+. However, it has been shown that this population is heterogeneous in terms of health problems. Therefore, our analysis was conducted in two subgroups—people in early and late old age. Moreover, the analysis of mortality trends using the joinpoint regression allowed to draw conclusions regarding changes in the fall-related mortality model which is an important contribution to the field and provides the novelty of the research. The results of the study can be used to set strategic directions for health policy regarding fall prevention.

The justification of the study is the fact that falls of the elderly constitute a serious geriatric, psychiatric, social, and economic problem. These may cause significant morbidity and mortality. Falls can also threaten the independence of older people and may be responsible for an individual’s loss of independence. It is a growing socioeconomic problem which may add extra burden to the healthcare—especially with the ageing of populations of many countries, including Poland. This raises a need to conduct research in this area.

The aim of this study was to assess mortality trends due to falls in the early (65–74 years) and late (75+) old age groups in Poland between the years 2000 and 2020. Specific objectives of the study included assessment of changes in mortality due to falls in two age groups distinguished by sex, using CDR and SDR, taking into account the rate and significance of changes in trend direction in the analyzed period. 

## 2. Materials and Methods

This study used a database of deaths of all Polish residents between 2000 and 2020, included in death certificates obtained from the Central Statistical Office in Poland. From this database, all deaths due to falls (according to the International Statistical Classification of Diseases and Health Related Problems—Tenth Revision—ICD-10, coded as W00–W19) and two age groups of the deceased were distinguished: early old age (65–74 years) and late old age (75+). The total number of deaths due to falls of people aged 65 years and older between the years 2000 and 2020 was 61,994 (10,560 in early old age and 51,434 in late old age).

Crude death rates (CDRs) and standardized death rates (SDRs) were calculated according to the following formulas:$$CDR=\frac{k}{p}\times 100,000$$
where *k*—number of deaths; *p*—population size.
$$\mathrm{SDR}=\frac{\sum_{i=1}^{N}\frac{ki}{pi}wi}{\sum_{i=1}^{N}wi}$$
where *k_i_* is the number of deaths in this i-age group, *p_i_* is population size of this i-age group, *w_i_* is the weight assigned to this i-age group, resulting from the distribution of the standard population, and *N*—number of the age groups.

The standardization procedure was performed with the use of a direct method, in compliance with the European Standard Population, updated in 2012 [22]. The Revised European Standard Population is an unweighted average of individual populations of EU-27 and EFTA countries in each 5-year age band (with the exception of people under 5 years of age and 85 years or older).

An analysis of time trends was carried out with joinpoint models and the Joinpoint Regression program, a statistical software package developed by the U.S. National Cancer Institute for the Surveillance, Epidemiology and End Results Program [23].

The joinpoint regression model is an advanced version of linear regression y = bx+a, where b is the slope coefficient, a is the y-intercept, y = ln(z), z is the measure evaluated in the study (SDR), and x is the calendar year. Time trends were determined with the use of segments joining in joinpoints, where trend values significantly changed (*p* < 0.05). To confirm whether the changes were statistically significant, the Monte Carlo permutation method was applied.

In addition, the authors also calculated annual percentage change (APC) for each segment of broken lines and average annual percentage change (AAPC) for the whole study period with corresponding 95% confidence intervals (CI).

Annual percent change is one way to characterize trends in death rates over time, and it was calculated according to the following formula:$$APC=100\times ({exp}^{b}-1)$$
where *b*—the slope coefficient.

Average annual percent change (AAPC) is a summary measure of the trend over a prespecified fixed interval. It allows us to use a single number to describe the average APCs over a period of multiple years. It is valid even if the joinpoint model indicates that there were changes in trends during those years. It is computed as a weighted average of the APCs from the joinpoint model, with weights equal to the length of the APC interval [24].
$$AAPC=\left\{exp\left(\frac{\sum w_{i}b_{i}}{\sum w_{i}}\right)-1\right\}\times 100$$
where *b_i_*—the slope coefficient for each segment in the particular range of years, and *w_i_*—the length of each segment in the range of years.

## 3. Results

The number of deaths due to falls in men in early old age (65–74 years) increased from 310 in 2000 to 501 in 2020 (Table 1).

The crude death rate (CDR) in 2000 was 25.3 per 100,000 men in this age group. Between 2000 and 2009, CDRs remained stable (Table 1, Figure 1).

Between the years 2009 and 2012, a statistically insignificant increase was observed, while after the year 2012, CDRs were significantly decreasing at an average annual rate (APC) of −2.3%. As a result of these changes, the CDR value in 2020 was 25.9 per 100,000 men. Similar trends were observed for standardized death rates (SDRs). After a slight, statistically insignificant decrease in the SDR values between 2000 and 2009 and a statistically insignificant increase between 2009 and 2012, SDRs significantly decreased between 2012 and 2020 (APC = −2.4%) (Table 2, Figure 2). As a consequence, the SDR value increased from 25.9 per 100,000 men in 2000 to 26.2 in 2020 (Table 1).

Among men in late old age (75+), the number of deaths due to falls increased from 667 in 2000 to 1051 in 2020. Per 100,000 men, the CDR value in 2000 was 118.5 (Table 1). Between 2000 and 2005, CDR values decreased at an average annual rate of −5.9% (*p* < 0.05), while after 2005 they began to increase at a rate of 1.3% (*p* < 0.05) and reached a value of 115.2 in 2020 (Table 2, Figure 1).

SDR values due to falls in the late old age male group, after a rapid decline between 2000 and 2009 (APC = –4.5%; *p* < 0.05), were characterized with statistically insignificant periodic increases (between 2009–2013 and 2016–2020) and decreases (between 2013–2016) (Table 2, Figure 2). As a result, the SDR value decreased from 160.6 in 2000 to 118.1 in 2020 (Table 1).

There were 240 deaths due to falls in the early old age female group in 2000, and 204 deaths in 2020. The CDR values in 2000 and 2020 were 13.9 and 8.2 per 100,000 women aged 65–74 years, respectively (Table 3).

Between 2000 and 2007, CDRs decreased at an average annual rate of −6.8% (*p* < 0.05). A very small and statistically insignificant increase of 0.2% was noted after 2007 (Table 2, Figure 1). The SDR value in the early old age group of women decreased from 14.0 in 2000 to 6.4 in 2007 (APC = −7.2%; *p* < 0.05) (Table 2, Table 3). After 2007, SDR values were slightly but insignificantly increasing at a rate of 0.8%, and in 2020 they reached the value of 8.3 per 100,000 (Table 3, Figure 2).

There were 1831 deaths in women in late old age in 2000 and 1992 deaths in this age group in the year 2020 (Table 3). The CDR value in the year 2000 was 151.5 per 100,000 women aged 75 years and older. Between the years 2000 and 2008, CDRs decreased at an average annual rate of −6.9% (*p* < 0.05). After 2008, they began to increase at an average annual rate of 1.9% (*p* < 0.05), reaching the value of 111.6 per 100,000 in 2020 (Table 3, Figure 1). A higher than twofold decrease in the SDR value in the group of women aged 75 years and older was observed between 2000 and 2009 (188.9 in 2000 and 89.8 in 2009) (Table 3, Figure 2). The average rate of decline between 2000 and 2009 was −7.5% (*p* < 0.05). There was a very slight, statistically insignificant decrease in the SDR value between the years 2009 and 2020 (APC = −0.1%). In 2020, the SDR value was 98.0 per 100,000 women in the late old age group.

## 4. Discussion

Falls are a serious medical, psychosocial, and economic problem. They can gradually reduce self-dependence, which in turn considerably worsen the quality of life [25,26].

The change in the age structure of the population, observed in most countries of the world, is the main factor of the increase in the absolute number of fatal falls. The number of people aged 65 years and older in Poland is steadily increasing at an alarming rate. In 2000, the number of men aged 65–74 years was 1,224,355, while the number of men aged over 75 years was 562,705. In 2020, these numbers were 1,932,697 and 912,463, respectively (an increase by approximately 58% and 62%). In the year 2000, there were 1,730,079 women aged 65–74 years and 1,208,638 women aged 75 years or older. In 2020, these numbers increased by approximately 44% and 48%, respectively. There were 2,489,826 and 1,784,999 women in early old age and in late old age.

As the elderly population increases, the number of deaths from causes which are dominant in the older age group increases. These causes of death include falls. A total of 76% of the total number of deaths due to falls, registered in Poland in 2020, occurred in people aged 65 years and older (15% in the 65–74 age group and 62% in the 75 years and older age group) [27].

The increase in crude death rates in the late old age group observed in our study (since 2005 in the group of men and since 2008 in the group of women) is mainly related to the aforementioned changes in the age structure of the Polish population. Age-standardized death rates in the group of women aged 75 years and older remained relatively stable after 2009, while in the group of men aged 75 years and older they were subject to periodic, statistically insignificant increases and decreases. The lack of significant improvement in mortality rates due to falls in the Polish elderly population observed since 2009, despite a favorable trend in the first decade of the analysis, may be related to an increase in the dynamics of growth in the trend of the oldest people (i.e., those aged over 80), particularly noticeable in the last decade (from 6.4% to 8.5%). Indeed, mortality rates due to falls increase exponentially with age in both sexes, reaching their highest values at the age of 85 and older. This can be attributed to age-related progressive degeneration of cognitive, sensory, and physical functions, as well as an increase in the number of comorbidities and the phenomenon of polyphagia [28].

With regards to the profile of people burdened with the highest risk of falling, considering all groups of factors, it can be concluded that elderly women, slim, with weakened muscles and gait disturbances, affected by multimorbidity, taking more than four medications per day and with a history of falls are at the highest risk of falling [29]. Women are three times more likely to experience a fall than men [30]. However, an analysis of mortality in the subpopulation of people experiencing falls reveals that the male population, rather than the female one, faces an increased risk of death due to this cause. This observation was confirmed in this study for both the early and late age groups. This is due to several factors. Firstly, men suffer from more comorbidities than women of the same age [31]. In addition, this is a result of different circumstances of the fall and kind of sustained injuries, which, as O’Neill points out in his study, is determined by gender [32]. Men are more likely to fall down outdoors, during acute episodes of the disease, hereby being more likely to sustain head and thoracic injuries, which increases mortality rates due to this cause [33].

Worldwide mortality trends due to falls are various in various areas. This can be explained by differences between countries both in terms of demographic structure (including race) and in terms of activity patterns of the older population [34]. In the year 2000, in which the authors of this study initiated their research, the average SDR values per 100,000 population in European Union countries, the WHO European Region, and in Poland were 50.9, 39.5, and 77.9, respectively. In all cases, a decrease in these values was observed by 2020—by 19.6%, 14.2%, and 44.7%, respectively [16]. In 2000, the highest mortality due to falls among European countries was observed in Hungary, i.e., 181.93 per 100,000 population. However, over a period of 20 years, a favorable trend was observed in this respect, and in the year 2020, the SDR value was almost three times lower (69.0). Over the analyzed period, the trends improved in most European countries. The greatest dynamics were noted in the first decade, which corresponds to the results of our analysis. In contrast, unfavorable mortality trends from this cause were particularly observed in the Netherlands (an increase from 7.9 in 2000 to 100.68 in 2020), Slovenia (an increase from 95.18 to 159.29), and Croatia (an increase from 91.78 to 111.96) [20]. Data obtained from the Center for Disease Control and Prevention revealed that in the year 2018, standardized death rates in the United States due to falls in people aged 65 years and older was 64 deaths per 100,000 elderly people, and has increased by approximately 30% since 2009. The increase was observed in 30 states and in the District of Columbia. The rate was growing most rapidly in the population aged 85 years and older (approximately 4% per year) [34,35,36]. In Poland, favorable mortality trends due to falls in the late old age group also started to decrease, and this negative trend began in the year 2009. Studies conducted in China between 2013 and 2020 also revealed unfavorable fall-related mortality trends in the population aged 65 years and older during the study period, particularly among women aged 85 years and older, where the rate of increase expressed as average percentage change (APC) was the highest [37].

Prognoses of the National Institute of Geriatrics, Rheumatology and Rehabilitation indicate that by 2050, the number of falls among the elderly in Poland will have almost doubled, which will result in nearly 2.3 million cases [38]. However, research shows that effective public health interventions prevent falls and their complications. From a public health perspective, concerted action should be taken to reduce the number of falls among the elderly in order to reduce fall-related injuries and complications. Multifaceted strategies aimed at reducing risk factors of the incidence of falls among people at high risk should be implemented. Specific interventions might include the development of policies to prevent falls in long-term care facilities and public places and education sessions on how to prevent falls. The key is to identify people at risk of falling and to refer them to local programs or resources. Prevention of falls must span the spectrum of ages and health states within the older population and address the diversity of causes of falls without unnecessarily compromising quality of life and independence [39]. Preventive actions usually involve improving safety in the home environment and the immediate environment of the elderly person, making a proper diagnosis and implementing an appropriate therapy of current diseases, monitoring pharmacotherapy, initiating exercises to improve balance and motor coordination, providing assistive equipment, and educating the patient and his/her relatives [40,41]. Taking comprehensive and personalized preventive measures can reduce the number of falls in the elderly by as much as 40 to 60% [42,43], which confirms that there is a need to carry out in-depth research in this area. The above measures might also enable reducing the health, social, and economic impacts of this phenomenon in the subpopulation of elderly people as well as in the general population.

## 5. Conclusions

The number of elderly people in Poland is increasing rapidly, which entails an increase in the number of fatal falls. However, falls are still not considered a public health problem, and this negligence is evidenced by the lack of comprehensive epidemiological data on detailed assessment of the situation and trends in this area. This raises a need to conduct in-depth research in this area, as assessment of mortality trends due to falls in different subpopulations can help to identify needs and implement appropriate prevention programs for specific target groups. In view of current demographic trends, fall prevention should be a priority in healthcare in Poland, and particular attention should be paid to tailoring interventions to cohorts of the oldest population and to those who are at the highest risk of consequences of falls.

## Figures and Tables

**Figure 1 ijerph-20-05073-f001:**
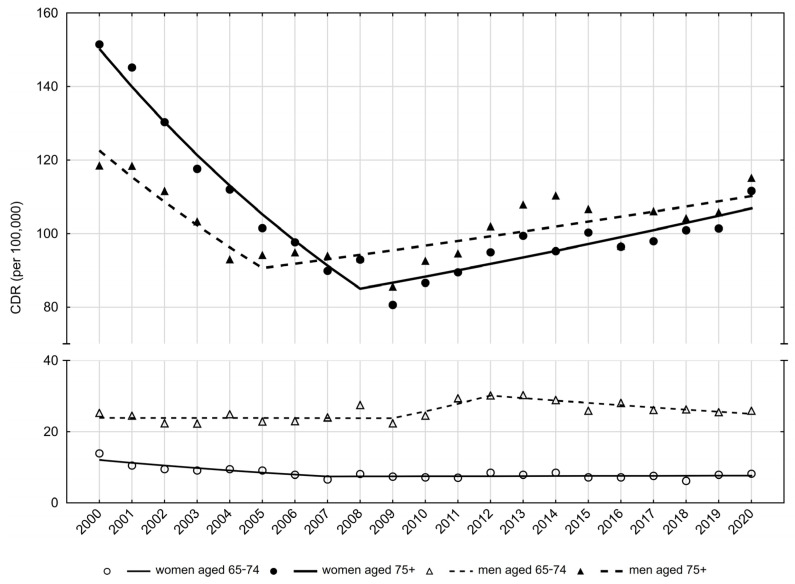
Trends of crude death rates due to falls in people aged 65–74 years and 75 years and older in Poland in 2010–2020. Source: own calculations.

**Figure 2 ijerph-20-05073-f002:**
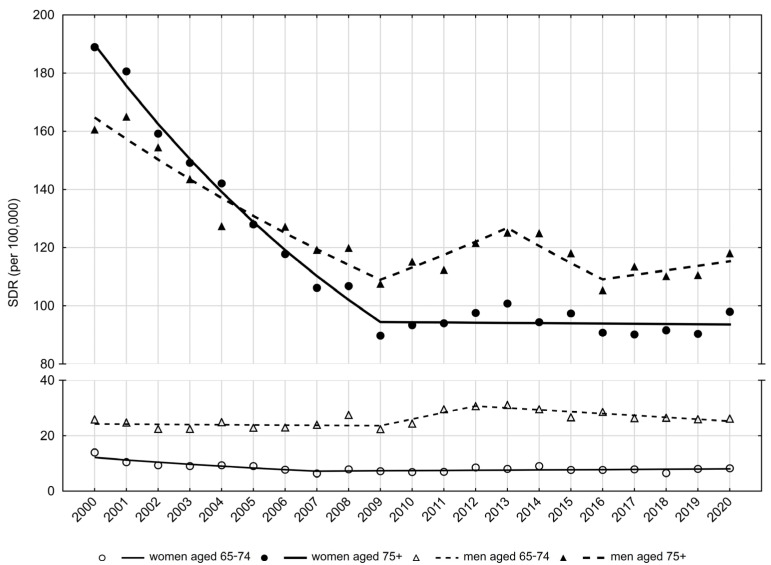
Trends of standardized death rates due to falls in people aged 65–74 years and 75 years and older in Poland in 2010–2020. Source: own calculations.

**Table 1 ijerph-20-05073-t001:** Number of deaths, crude death rates (CDRs), and standardized death rates (SDRs) due to falls in men aged 65–74 years and 75 years and over in Poland in 2000–2020.

Year	65–74 Years			75 Years and Over		
	Number of Deaths	CDR	SDR	Number of Deaths	CDR	SDR
2000	310	25.3	25.9	667	118.5	160.6
2001	303	24.5	24.8	691	118.4	165.0
2002	279	22.4	22.5	672	111.6	154.5
2003	277	22.3	22.5	652	103.3	143.6
2004	307	24.9	24.9	617	93.0	127.4
2005	279	22.9	22.9	659	94.2	128.4
2006	276	23.0	23.0	695	94.9	127.2
2007	282	24.0	24.0	714	94.0	119.3
2008	319	27.5	27.5	729	93.3	119.5
2009	256	22.4	22.4	685	85.6	107.6
2010	279	24.5	24.4	759	92.6	115.2
2011	346	29.4	29.6	797	94.6	112.4
2012	373	30.2	30.7	879	102.0	121.6
2013	395	30.3	31.1	947	107.9	125.2
2014	399	28.9	29.6	988	110.4	125.0
2015	378	25.9	26.7	968	106.7	118.1
2016	436	28.1	28.6	877	96.7	105.4
2017	431	26.1	26.4	974	106.1	113.5
2018	461	26.3	26.5	958	104.2	110.2
2019	471	25.5	26.0	978	105.8	110.6
2020	501	25.9	26.2	1051	115.2	118.1

Source: own calculations.

**Table 2 ijerph-20-05073-t002:** CDR and SDR time trends due to falls in Poland in the years 2000–2020—joinpoint regression analysis.

	Number of Joinpoints	Years	APC (95% CI)	AAPC (95% CI)
CDR				
Men aged 65–74 years	2	2000–2009	0.0 (−1.7; 1.6)	0.2 (−0.4; 2.9)
		2009–2012	8.2 (−9.6; 29.5)	
		2012–2020	−2.3 * (−4.2; −0.4)	
Men aged 75 years and older	1	2000–2005	−5.9 * (−9.1; −2.6)	−0.5 (−1.4; 0.4)
		2005–2020	1.3 * (0.7; 2.0)	
Women aged 65–74 years		2000–2007	−6.8 * (−10.2; −3.2)	−2.3 * (−3.7; −0.8)
		2007–2020	0.2 (−1.2; 1.7)	
Women aged 75 years and older	1	2000–2008	−6.9 * (−7.9; −5.8)	−1.7 * (−2.2; −1.2)
		2008–2020	1.9 * (1.3; 2.5)	
SDR				
Men aged 65–74 years	2	2000–2009	−0.3 (−1.9; 1.4)	0.2 (−2.4; 3.0)
		2009–2012	9.2 (−9.0; 31.2)	
		2012–2020	−2.4 * (−4.3; −0.4)	
Men aged 75 years and older	3	2000–2009	−4.5 * (−5.6; −3.4)	−1.8 (−3.9; 0.4)
		2009–2013	3.9 (−2.5; 10.6)	
		2013–2016	−4.9 (−16.1; 7.9)	
		2016–2020	1.4 (−2.6; 5.5)	
Women aged 65–74 years	1	2000−2007	–7.2 * (−10.7; −3.5)	−2.0 * (−3.6; −0.5)
		2007–2020	0.8 (−0.7; 2.4)	
Women aged 75 years and older	1	2000–2009	−7.5 * (−8.3; −6.6)	−3.5 * (−4.0; −3.0)
		2009–2020	−0.1 (−0.8; 0.6)	

Source: own calculations. * *p* < 0.05.

**Table 3 ijerph-20-05073-t003:** Number of deaths, crude death rates (CDRs) and standardized death rates (SDRs) due to falls in women aged 65–74 years and 75 years and over in Poland in 2000–2020.

Year	65–74 Years			75 Years and Over		
	Number of Deaths	CDR	SDR	Number of Deaths	CDR	SDR
2000	240	13.9	14.0	1831	151.5	188.9
2001	183	10.5	10.5	1805	145.2	186.0
2002	167	9.5	9.4	1664	130.3	159.2
2003	160	9.1	9.1	1561	117.6	149.2
2004	166	9.5	9.4	1546	112.0	142.1
2005	157	9.1	9.1	1462	101.5	128.0
2006	133	7.9	7.8	1457	97.6	117.8
2007	109	6.6	6.4	1384	89.9	106.2
2008	132	8.1	7.9	1464	92.9	106.8
2009	119	7.4	7.2	1299	80.6	89.8
2010	113	7.2	6.9	1433	86.6	93.4
2011	114	7.1	7.0	1517	89.5	94.0
2012	142	8.5	8.6	1638	94.9	97.6
2013	137	7.9	8.1	1742	99.4	100.8
2014	154	8.5	9.1	1696	95.2	94.4
2015	137	7.2	7.7	1805	100.3	97.4
2016	146	7.2	7.7	1749	96.4	90.8
2017	163	7.6	7.9	1772	97.9	90.2
2018	139	6.2	6.5	1820	100.9	91.6
2019	187	7.9	8.1	1830	101.4	90.4
2020	204	8.2	8.3	1992	111.6	98.0

Source: own calculations.

## Data Availability

The data presented in this study are available on request from the corresponding author.

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
