# Peer review of "Mortality Trends due to Falls in the Group of People in Early (65–74 Years) and Late (75+) Old Age in Poland in the Years 2000–2020"

_ijerph, 2023, doi:10.3390/ijerph20065073_

Round 1

Reviewer 1 Report

I do not consider the manuscript suitable for publication in this journal. The manuscript addresses a topic with a limited population that may not reflect the real trends of this problem in different countries and continents. For the manuscript to become more relevant, the authors should investigate populations from other countries and even continents. Thus, we would have statistical data that are more significant and closer to reality.

Author Response

Dear Reviewer,

Thank You for the opinion, but we cannot agree with it.

“IJERPH covers a broad spectrum of important topics which are relevant to environmental health sciences and public health protection”. The problem covered by the study, constituting the subject of our article, is an important issue of public health, especially in the face of aging societies, and fits the aim and scope of the journal. The results presented in the manuscript concern the whole country (38 mln residents), not a selected region, city or province. A large number of published scientific articles concern research conducted within one country. Reference to data from other countries of the world is then usually included in the discussion section. In our article, one paragraph in the discussion contains a reference to data on mortality due to falls for selected countries of the world. The study results presented in the manuscript constitute the effect of analyses of data on all deaths of Polish residents aged 65+ in the years 2000-2020. This comprehensive analysis spanning the first two decades of the 21st century poses, in the authors' opinion, an important contribution to the literature and the field of public health.

Reviewer 2 Report

The authors studied the trend of mortality of elderly people due to falls in Poland, which is a very important issue and deserves our attention. But there are still three recommendations for the authors.

First, we still do not know anything new about the causes of falls in the elderly in this paper. Is there a possible explanation for why the data trend in Poland follows this pattern?

Second, is the direct cause of death of elderly people who die from a fall? Or did these elderly people die due to other complications caused by fall?

Third, what measures can be taken by the public health sector to avoid deaths due to fall.

Author Response

Dear Reviewer,

we are very grateful for insightful analysis of our work. We are also convinced that all the amendments will contribute to improve the quality of our manuscript.

Manuscript has been redrafted in accordance with the Reviewer's suggestions. Below, we refer in detail to the individual comments.

  1. The introduction of the article includes a short fragment on the causes and risk factors of falls among the elderly (line 55-60). In the discussion section, an attempt to link mortality trends due to falls with the most important variables, such as age, sex, multimorbidity was made. However, there are no data on the prevalence of other fall risk factors in the population (also due to their non-specificity), which could be directly attributed to their causality and mortality due to this cause. The death certificates that constituted the database for our study do not contain information on the cause of the fall when death is caused by it. So we have been unable to relate these facts. The aim of the study was to assess trends in mortality due to falls in order to assess the burden of this phenomenon on the population. However, an analysis of the causes would require a separate study, using a completely different methodology.
  2. Data on deaths by cause are compiled according to the underlying cause of death according to the rules adopted by the World Health Assembly (WHA) regarding the selection of a single cause or condition, from death certificates, for the routine tabulation of mortality statistics for standardization. It was agreed by the Sixth Decennial International Revision Conference 1948 (for ICD) that the cause of death for primary tabulation should be designated the underlying cause of death. WHO has defined the ‘underlying cause of death’ as follows: the disease or injury which initiated the train of morbid events leading directly to death, or the circumstances of the accident or violence which produced the fatal injury" [https://www.who.int/standards/classifications/classification-of-diseases/cause-of-death].
  3. The relevant fragment has been added to the last part of the Discussion section to extend fragment about role of public health  (line 326-335).

Reviewer 3 Report

Dear Author

This is a very interesting article kindly see my minor comments on the attached PDF file.

Author Response

Dear Reviewer,

we are very grateful for insightful analysis of our work. We are also convinced that all the amendments will contribute to improve the quality of our manuscript.

Manuscript has been redrafted in accordance with the Reviewer's suggestions. Below, we refer in detail to the individual points of comment.

  1. The abbreviation WHO has been added.
  2. A reference has been added.
  3. A reference has been added. Some of the environmental risk factors have been listed (line 58-59).
  4. Coma instead of add has been insert.
  5. The abbreviation GBD has been added.
  6. Instead of “the study” – “this study” has been inserted.
  7. The unnecessary paragraph has been removed. It was posted there by mistake. Thank You for pointing it out. The ethical statement has been provided in appropriate place of the manuscript (line 363-365).

Kind regards

Authors

Reviewer 4 Report

In this study, authors studied about mortality trend due to fall in early and late age in Poland during 2000-2020. I urge the authors to address the following suggestions:

1. Please do not use abbreviations without mentioning their full words where firstly used, like SDR, CDR in abstract even.

2. There are many grammatical mistakes in the manuscript. Please revise it thoroughly.

3. Please cite some latest references of the respective journal.

4. Please mention the novelty statement clearly in the manuscript.

5. Please provide the aims and objective of the study.

6. Rationale of the study is missing. Please provide it.

Author Response

Dear Reviewer,

we are very grateful for insightful analysis of our work. We are also convinced that all the amendments will contribute to improve the quality of our manuscript.

Manuscript has been redrafted in accordance with the Reviewer's suggestions. Below, we refer in detail to the individual points of comment.

  1. Abbreviations have been clarified with the first use.
  2. Manuscript has been revised. At the request of the Reviewer, we can present the file in the mode of tracking changes made during proofreading. This has not been done at this stage to make the file readable by all reviewers.
  3. Some latest references of the respective journal have been cited.
  4. Novelty statement has been mentioned in the Introduction section (line 92-100).
  5. Specific objectives have been added apart from main goal (line 108-111).
  6. Rationale of the study has been provided in the Introduction section (line 101-106).

Kind regards

Authors

Round 2

Reviewer 1 Report

My opinion on the manuscript remains the same as the first round of review.

Author Response

Dear Reviewer, thank You very much for Your opinion.

However, in accordance with the recently presented assumption, we maintain the opinion that the issue discussed in the article is important from the point of view of preventing falls and can be used to reduce mortality as well as to reduce the negative consequences of falls which are very serious problem of public health.

Authors

Reviewer 4 Report

Authors have addressed the comments adequately.

Author Response

Dear Reviewer,

thank You very much for Your review and for accepting all made changes.